# Deep Learning-Based Hardness Prediction of Novel Refractory High-Entropy Alloys with Experimental Validation

Uttam Bhandari [1,*], Congyan Zhang [1], Congyuan Zeng [2], Shengmin Guo [2], Aashish Adhikari [3] and Shizhong Yang [1,*]

[1] Department of Computer Science, Southern University and A&M College, Baton Rouge, LA 70813, USA; congyan_zhang@subr.edu

[2] Department of Mechanical and Industrial Engineering, Louisiana State University, Baton Rouge, LA 70803, USA; czeng8@lsu.edu (C.Z.); sguo2@lsu.edu (S.G.)

[3] School of Electrical Engineering and Computer Science, Oregon State University, Corvallis, OR 97331, USA; adhikara@oregonstate.edu

\* Correspondence: uttam.bhandari@sus.edu (U.B.); shizhong_yang@subr.edu (S.Y.)

**Abstract:** Hardness is an essential property in the design of refractory high entropy alloys (RHEAs). This study shows how a neural network (NN) model can be used to predict the hardness of a RHEA, for the first time. We predicted the hardness of several alloys, including the novel $C_{0.1}Cr_3Mo_{11.9}Nb_{20}Re_{15}Ta_{30}W_{20}$ using the NN model. The hardness predicted from the NN model was consistent with the available experimental results. The NN model prediction of $C_{0.1}Cr_3Mo_{11.9}Nb_{20}Re_{15}Ta_{30}W_{20}$ was verified by experimentally synthesizing and investigating its microstructure properties and hardness. This model provides an alternative route to determine the Vickers hardness of RHEAs.

**Keywords:** high entropy alloys; neural networks; hardness-prediction; microstructure

## 1. Introduction

Refractory high entropy alloys (RHEAs) are designed by adding high melting point elements such as Cr, Mo, Nb, Ta, Ti, V, W, Re, W, Ru, Zr, Rh, Os, Ir, and Hf. RHEAs have excellent properties, such as a high hardness and high-temperature softening resistance [1–4]. Thus, application of RHEAs can be very promising candidates in the field of aerospace industry. Many studies of RHEAs such as TaNbHfZrTi, TiZrNbMoV, NbMoTaW, VNbMoTaW, $Ti_2ZrHf_{0.5}MoNbx$, and $Al_{0.5}CoCrCuFeNi$ have shown excellent yield strength both at room and high temperatures [5–10]. However, the poor ductility at room temperature of RHEAs limits their industrial application. One study has shown that doping elements such as C, B, N, O, Al, Si with RHEAs could improve the mechanical properties of RHEAs and enhance their applications [11–18]. The addition of 0.1 atomic percentage (at.%) of C and 0.3 at.% of C in $Mo_{0.5}NbHf_{0.5}ZrTi_{10}$ has improved the plasticity and increased the compressive stress [19]. Therefore, doping with low concentrations of non-metallic elements is very important in RHEAs for improving the mechanical properties. However, predicting the mechanical properties of doped RHEAs by the first principles method is very complex. Moreover, experimental techniques are costly and tedious. Therefore, machine learning (ML) could be a promising tool to predict the mechanical properties of new complex RHEAs.

ML approaches have been used to predict the crystal structures and properties of materials [20–22] with convincing results. Islam et al. [23] trained neural network (NN) models that can analyze the phase of multi-principal element alloys (MPEAs) with 118 data samples. An NN model is a supervised machine learning prediction model that is inspired by the functionalities of the human brain. Wen et al. [24] developed a material design strategy using ML for predicting the desired properties of high entropy alloys (HEAs). George et al. [25] also utilized a machine learning model which can predict the

elasticity of high entropy alloys and included experimental validation. These studies have shown that ML methods are capable and reliable in discovering new RHEAs and predicting their phases. However, many studies focus on phase formation predictions, and a few on predicting the mechanical or functional properties of complex RHEAs.

In this study, the Vickers hardness of several complex RHEAs including $C_{0.1}Cr_3Mo_{11.9}Nb_{20}Re_{15}Ta_{30}W_{20}$ were predicted by utilizing an NN model. We experimentally synthesized $C_{0.1}Cr_3Mo_{11.9}Nb_{20}Re_{15}Ta_{30}W_{20}$, and the phase, microstructure, and Vickers hardness were studied. The measured Vickers hardness was found to be consistent with the NN model prediction.

## 2. Materials and Methods

### 2.1. Computational Methods

The features of the alloys and the corresponding target hardness values used in this study were collected from several previous publications [6,19,26,27]. In this study, we only considered the average hardness value of HEAs present in the reference papers and did not consider the hardness value that changed with the applied load. Although a total of 380 experimental records of measured hardness were collected from the literature, the alloys that contained unwanted elements were left out, reducing the number of total available samples to 128.

A supervised machine learning task performs several steps before the actual prediction is made. An NN model requires features to train and thus make informed predictions. In this particular task, 22 relevant features that were initially picked based on the domain knowledge of which features could be helpful to predict Vickers hardness consisted of the percentage of each element in the alloy, i.e., % of Cr, % of Hf, % of Mo, % of Nb, % of Ta, % of Ti, % of Re, % of V, % of W, % of Zr, % of Co, % of Ni, % of Fe, % of Al, % of Mn, % of Cu, % of C, the entropy of the alloy ($\Delta S_{mix}$) [28], the bulk modulus ($B$), the shear modulus ($G$), the valence electron concentration ($VEC$) [29], and the melting temperature ($T_m$). The HEA features were calculated using the rule of mixtures.

$$\Delta S_{mix} = -R \sum_{i=1}^{n} C_i ln C_i \tag{1}$$

$$B = \sum_{i}^{n} (C_i B_i) \tag{2}$$

$$G = \sum_{i}^{n} (C_i G_i) \tag{3}$$

$$T_m = C_i (T_m)_i \tag{4}$$

$$VEC = \sum_{i=1}^{n} C_i (VEC)_i \tag{5}$$

where $R$ is the ideal gas constant, $C_i$ the atomic percentages of the $i$th element, $B_i$ is the bulk modulus of the $i$th element, $G_i$ is the shear modulus of the $i$th element, $(T_m)_i$ is the melting point of the $i$th element, and $(VEC)_i$ is the valence electron concentration of the $i$th element.

In a typical supervised machine learning task, the dataset is split into training and testing sets before any feature engineering, visualization, analysis, or training is performed, so as to prevent any data leakage, i.e., the training should not be influenced by any information from the test data. After random shuffling, 90% of the total samples were allocated for training and validation and 10% for testing. Here, we explain the data preprocessing steps that were conducted on the training data before the NN model was trained. Firstly, the features that had zero or nominal variance were removed from the dataset. The methods adopted for feature selection and engineering in this work stem from a general class of feature engineering techniques [30]. ML models work on the basis of how close the samples are located to each other in the high-dimensional feature space. Features

that have nominal variance do not contribute much to differentiating among the samples, therefore it is better to remove such features before passing the data to an ML model to reduce space as well as time complexity. The removed feature was the "% of Re" in each sample, because all the samples in the training data had the same percentage of Re. Next, we removed the correlation among the features by two methods in succession, first, by the method of condition indices, and then by visually plotting the correlation among the features that successfully passed the condition index test.

A subset of the scatterplot of the features before the condition indices step is shown in Figure 1. There is clear association between that bulk modulus, valence electron concentration, entropy, and melting temperature, suggesting that these features are highly correlated. The calculated Pearson's correlation of coefficient [31] between the RHEA features of 128 samples' data is shown in Table 1.

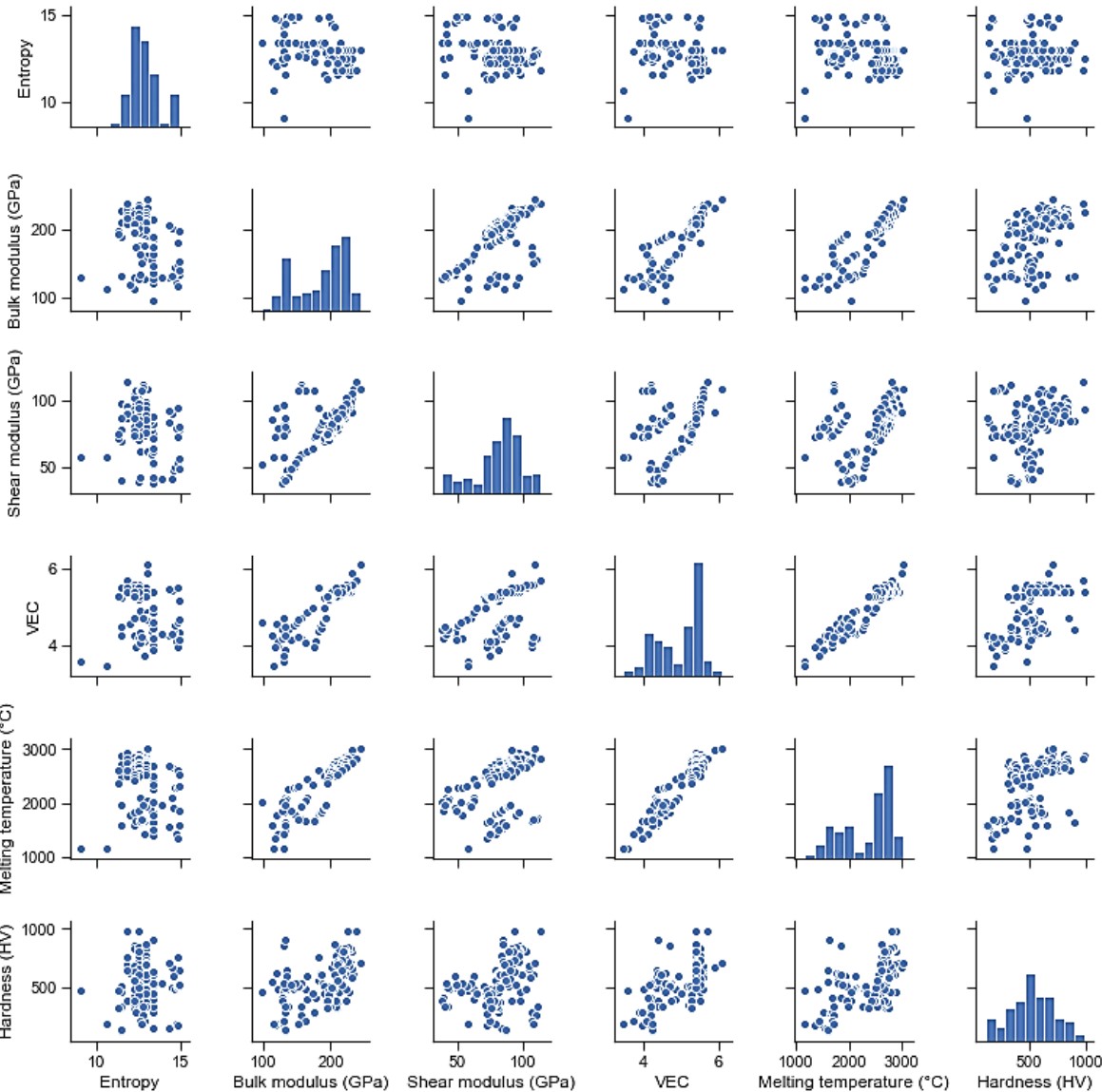

**Figure 1.** Scatterplot matrices of six features from the 128 samples data.

**Table 1.** The calculated Pearson's correlation coefficients which are related to different features of high entropy alloys (HEAs) in 128 data samples.

|  | $\Delta S_{\text{mix}}$ | $B$ | $G$ | $VEC$ | $T_m$ | HV |
|---|---|---|---|---|---|---|
| $\Delta S_{mix}$ | 1 | −0.24 | −0.25 | −0.16 | −0.12 | 0.044 |
| $B$ | −0.24 | 1 | 0.62 | 0.9 | 0.88 | 0.53 |
| $G$ | −0.25 | 0.62 | 1 | 0.48 | 0.37 | 0.32 |
| $VEC$ | −0.16 | 0.9 | 0.48 | 1 | 0.96 | 0.57 |
| $T_m$ | −0.12 | 0.88 | 0.37 | 0.96 | 1 | 0.56 |
| HV | 0.044 | 0.53 | 0.32 | 0.57 | 0.56 | 1 |

Table 1 shows that a correlation exists among the different features used. During feature removal, only those features that had minimal correlation with the target variable hardness were removed; this set of removed features included *VEC*, $T_m$, and **B**. This is a way to reduce the feature space complexity by losing minimal information from the dataset. Furthermore, visually plotting the scatterplots of the remaining features revealed some correlation among the % of Co, the % of Fe, and the % of Ni. Hence, the % of Co and the % of Fe were eliminated.

Next, Pandas library [32] was used to normalize the value of the features so that each feature came within a range of 0 and 1. Feature normalization allows all the features to scale to a similar scale, and thus makes training faster by balancing the length of the contours of the loss function across all dimensions of the parameters that are to be updated.

After that, the actual training steps began. The number of training samples is very small; therefore, k-fold cross-validation was chosen to obtain a model which performed the best on average among all the validation folds. Furthermore, because a neural network training comes with several hyperparameter choices, the hyperparameters were tuned using a randomized search. The hyperparameter grid used to sample different configurations of randomized hyperparameter search is as below:

$$\text{learning rate} = \{0.00005, 0.0001, 0.0003, 0.0005, 0.001, 0.003, 0.005\}$$

$$\text{weight decay for regularization} = \{1 \times 10^{-5}, 5 \times 10^{-5}, 1 \times 10^{-4}, 5 \times 10^{-5}, 1 \times 10^{-3}, 5 \times 10^{-3}, 1 \times 10^{-2}, 5 \times 10^{-2}, 1 \times 10^{-1}\}$$

$$\text{momentum} = \{0.5, 0.75, 0.99\}$$

$$\text{nesterov} = \{\text{True, False}\}$$

$$\text{number of nodes per layer} = \{14, 17, 21, 24, 28, 35\}$$

$$\text{number of maximum epochs to train} = \{50, 75, 100, 125, 150, 175, 200, 250, 300, 325, 350, 375, 400, 425, 450, 475, 500\}$$

$$\text{dropout probability per node} = \{0, 0.3, 0.5\}$$

$$\text{minibatch size} = \{1, 2, 4, 8, 16, 32, 64\}$$

$$\text{optimizers} = \{\text{torch.optim.SGD}\}$$

The training methodology followed a four-step procedure:

Step 1. Randomly selected 500 different configurations from the hyperparameter space.

Step 2. Trained 500 neural networks using all of these configurations with 20-fold cross validation.

(a) The purpose of this k-fold training was to find the most generalizable set of hyperparameter configurations among the randomly chosen 500 configurations.

Step 3. After obtaining the best set of hyperparameters, we discarded all the previously trained 500 * 20 models and trained a new neural network using:

(b)     all the training data;

(c)     the best set of hyperparameters obtained from Step 2;

(d)     no cross validation. (because we had already found the hyperparameters).

Step 4: In Step 2, all intermediate models were saved at the end of each epoch. This allowed us to monitor the performance of our model on the test set at each epoch. In essence, if the model performance diverged after the nth epoch, we could discard the training epochs after the nth epoch. This gave us the "best model" corresponding to this "best set of hyperparameter configuration".

The open-source libraries PyTorch, scikit-learn, and scorch were used to train the NN model and predict the hardness of the RHEAs.

The NN model consisted of three abstract regions, namely, the input layer, the hidden layers, and the output layer. The input layer takes in the features and passes them to the hidden layers. The output layer is the last layer of the NN model, which simply produces the output based on the information passed from the last hidden layer nodes. The following relationship can be used to calculate the output of each neuron ($a_j$):

$$a_j = \sum_{i=1}^{n} x_i W_{ij} + b_j \tag{6}$$

where $b_j$ are the bias terms and $W_{ij}$ are the weights provided for each input node. The value of $a_j$ proceeds through an activation function. Then, the activation function will define the output of this particular neuron. The type of activation function used in this NN model is leaky ReLU [33]. Figure 2 shows the NN model used in this work, which consisted of five hidden layers. Each layer consisted of different numbers of neurons. A neural network is a universal function approximator, and the larger the network, the more able it is to learn the nuances in the feature space. The weights and biases were randomly initialized before each training. The optimization criterion to optimize was mean square error (*MSE)* loss.

$$MSE = \frac{1}{n} \sum_{i=1}^{n} \left( Y_i - \hat{Y}_i \right)^2 \tag{7}$$

where $n$ is number of data points, $Y_i$ is the observed values, and $\hat{Y}_i$ is the predicted values.

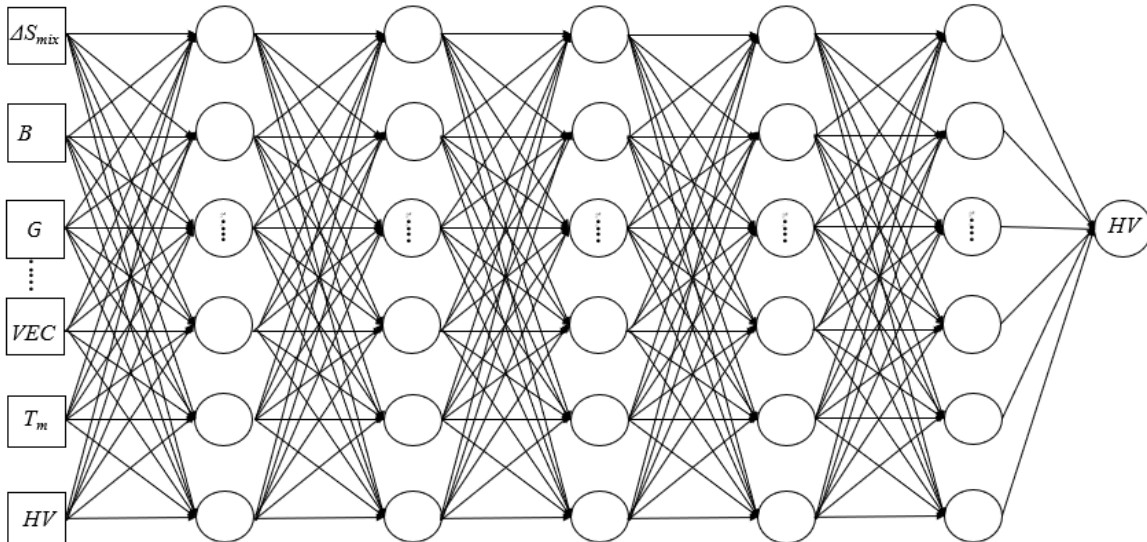

**Figure 2.** A schematic model of the neural network (NN) showing 5 hidden layers. The input consists of HEAs features as represented by square boxes on the left.

## 2.2. Experimental Methods

The $C_{0.1}Cr_3Mo_{11.9}Nb_{20}Re_{15}Ta_{30}W_{20}$ alloy was synthesized by vacuum arc melting method (MAM-1, Edmund Bühler, Bodelshausen, Germany) from the corresponding element mixture under a pure argon atmosphere. The powders of Cr, Mo, Nb, Re, Ta, W, and C were mixed inside a polystyrene mill jar for 15 min and cold pressed in a stainless steel mold using a pressure of 350 MPa and a dwelling time 30 s. The purity of each elements was more than 99.5 wt %. For the purpose of homogeneous distribution of elements, the melted ingots were flipped over and re-melted 4 times. After this, epoxy resin (a mixture of SamplKwick powder and SamplKwick liquid with the volume ratio 2:1) was applied to firmly wrap the solid ingots for easy handling. The cross section of the sample was cut using a BUEHLER low speed saw. Prior to the property tests, the cross section surface was mechanically ground with SiC papers with differing grit sizes, namely, 320, 600, 800, 1000, and 1200 mesh in sequence. After that, the ground cross section surface was polished using the MetaDi$^{TM}$ Supreme polycrystalline diamond suspension (1 μm), and, finally, rinsed with deionized water and dried in air. The crystal structure of the sample was identified using an X-ray diffractometer system (Empyrean, PANalytical) provided with equipped Cu K$\alpha$ radiation ($\lambda$ = 0.15406 nm). The 2θ scan range from 20–120 degrees was performed with a step size of 0.026 degrees. The microstructure and chemical compositions of the $C_{0.1}Cr_3Mo_{11.9}Nb_{20}Re_{15}Ta_{30}W_{20}$ were examined using a field emission scanning electron microscope equipped with second electron (SE) and energy dispersive spectroscopy (EDS) detector (Ametek. Model: APOLLO XL, Berwyn, PA, USA). Vickers hardness was measured with a digital micro hardness tester (Clark Instrument Model CM-802AT, Novi, MI, USA). Three testing loads were used, namely, 2000, 500, and 100 gf, and the dwell time was 15 s. The Vickers hardness (HV) of $C_{0.1}Cr_3Mo_{11.9}Nb_{20}Re_{15}Ta_{30}W_{20}$ was tested at five different positions on the sample to ensure the consistency of the results. The intervals between adjacent testing positions were three times larger than that of the indent diagonals, which avoided the effects of work hardening.

## 3. Results and Discussions

### 3.1. Machine Learning Results

The best set of hyperparameter configurations obtained were learning rate = 0.001, weight decay for regularization = 0.001, momentum = 0.99, optimizers = stochastic gradient descent, nesterov = True, number of nodes per layer = 35, number of maximum epochs to train = 325, dropout probability per node = 0, and minibatch size = 1. The learning curves of hardness prediction are shown in Figure 3. The *y*-axis represents the mean squared error and the *x*-axis represents the number of epochs. It can be seen from the curve that the NN model was able to learn from the training data and predict the hardness of the samples for the validation data with a gradual decrease in loss over several epochs of training, starting from 0.125 and flattening out at around 0.0675. The NN model was trained on a dataset of 128 HEAs. There were also several models that diverged during training. An example case is shown in Figure 4.

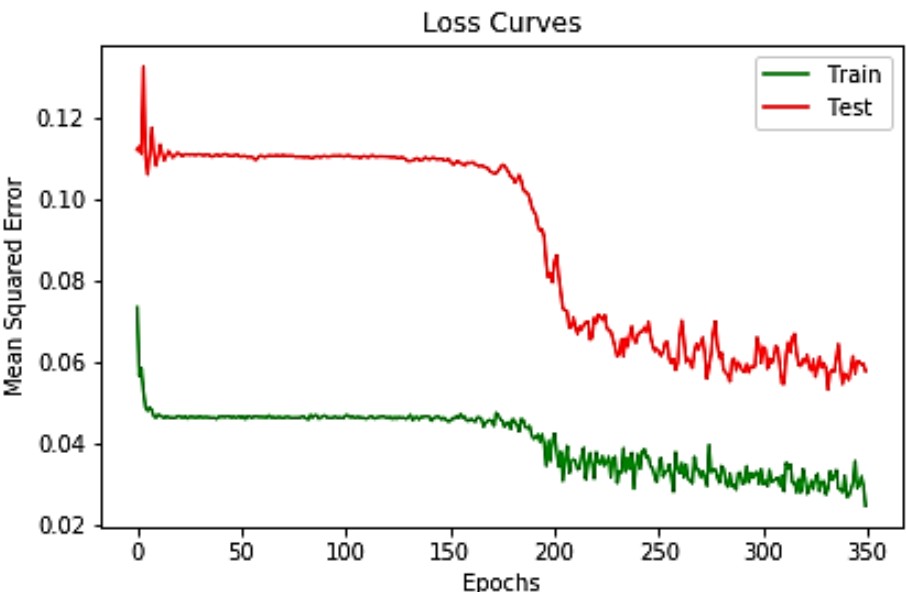

**Figure 3.** Learning curves of mean squared error as a function of the number of epochs. The red solid red curve represents the average 20-fold cross-validated loss scores on the validation fold, and the green solid curve corresponds to the 20-fold cross-validated loss score on the training data.

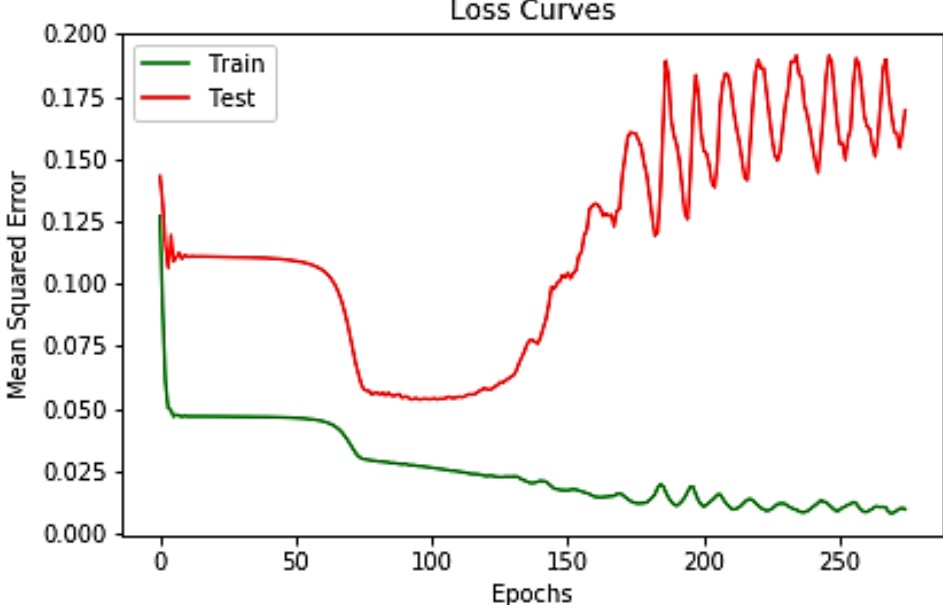

**Figure 4.** Mean squared error as a function of the number of epochs: a diverging model.

We also generated, for each test sample, the hardness predictions of all 500 models trained. Then, the uncertainty error bars were plotted using standard deviation as the metric, which are shown in Figure 5. As expected, there existed some variance among the predictions of all 500 models.

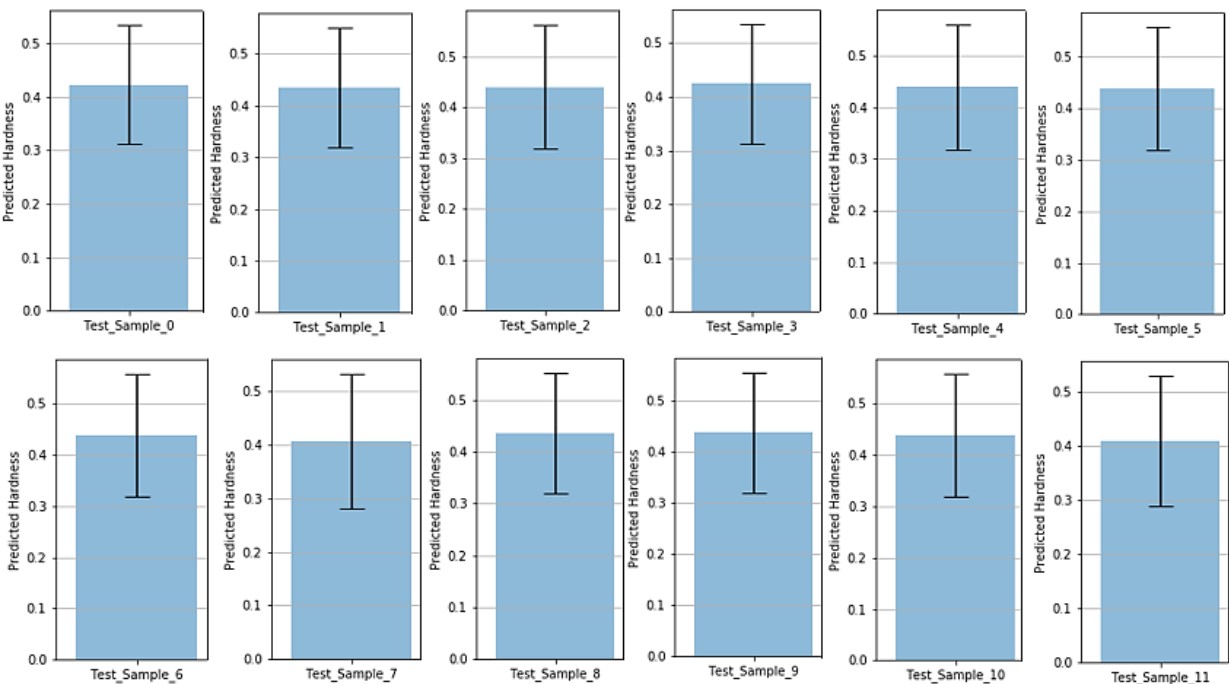

**Figure 5.** Error bars for the uncertainty among the trained models on test samples.

The R2 plots of Vickers hardness predicted by the NN on the training and test datasets are presented in Figure 6. As seen, the predictions have a trend along the y = x line. This indicates a good learnability for this task. After testing the model, the predicted hardness of $C_{0.1}Cr_3Mo_{11.9}Nb_{20}Re_{15}Ta_{30}W_{20}$ was found to be 690 HV. In Figure 6b, the training samples also digress from the ideal 45° line. This is because the model used for prediction was chosen to minimize the test loss and not the training loss: we chose the "early-stopped" model as the best model. This can also be observed in Figure 4. Table 2 contains the predicted and experimental Vickers hardness for the HEAs. As seen, the model predictions significantly differ for some samples. This variance is explained by the fact that the number of training samples was limited, and the training set did not contain enough samples from the region in the feature space that some test samples belonged to. This issue could be alleviated by using a large number of training samples from diverse regions in the feature space. We believe that if the NN model is retrained using a higher amount of HEA data, it can predict even better. Therefore, this limitation is not from the model but due to the unavailability of the data.

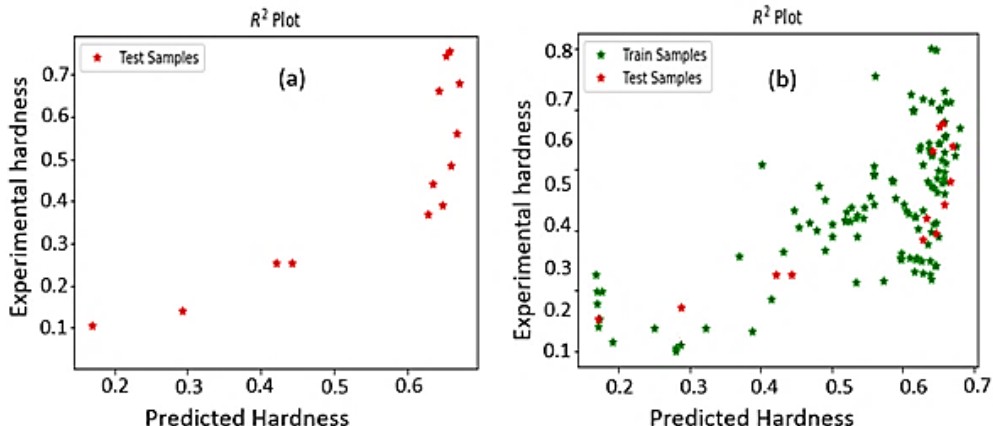

**Figure 6.** Hardness prediction by machine learning using the NN model for (**a**) the testing data and (**b**) both training and testing data. The red star symbol denotes the predicted hardness of the testing data, and the green star symbol denotes the predicted hardness of the training data.

**Table 2.** Experimental and NN model-predicted Vickers hardness of HEAs.

| Test Samples | Name of Alloys | Experimental Hardness (HV), and Reference | ML Prediction (HV) | Error % |
|---|---|---|---|---|
| | $C_{0.1}Cr_3Mo_{11.9}Nb_{20}Re_{15}Ta_{30}W_{20}$ (Nominal composition) | 601 (our work) | 695 | 15.6 |
| | $Cr_{1.6}Mo_{8.9}Nb_{20}Re_{15}Ta_{30}W_{20}$ (experimental composition) | 601 | 686 | 14.1 |
| 1 | $Hf_{21}Nb_{19.1}Ta_{20.1}Ti_{19.3}V_{23}Zr_{20.5}$ | 335 [34] | 500 | 49.2 |
| 2 | $Cr_{20}Mo_{20.2}Nb_{20.4}Ta_{20.6}V_{13}W_{20}$ | 704 [35] | 697 | 0.99 |
| 3 | $Nb_{25}Ti_{25}V_{25}Zr_{25}$ | 335 [36] | 481 | 43.5 |
| 4 | $Mo_{25.6}Nb_{22.7}Ta_{24.4}W_{27.3}$ | 454 [37] | 576 | 26.8 |
| 5 | $Cr_{20}Co_{19.3}Fe_{19.6}M_{17}Ni_{19.52}V_{4.6}$ | 151 [38] | 144 | 4.6 |
| 6 | $Al_{14.2}Mo_{22.2}Nb_{22.2}Ti_{21.5}V_{20.95}$ | 517 [39] | 598 | 15.6 |
| 7 | $Mo_{30}Nb_{10}V_{20}Ta_{20}W_{20}$ | 770 [27] | 686 | 10.9 |
| 8 | $Mo_{21.7}Nb_{20.6}Ta_{15.6}V_{21}W_{21.1}$ | 535 [37] | 687 | 28.4 |
| 9 | $Co_{12.9}Fe_{10.8}Ni_{10.8}Cu_{40.1}Al_{16.2}Si_{9.2}$ | 682 [40] | 762 | 11.7 |
| 10 | $Cr_{16.67}Co_{25.81}Ni_{25.81}Fe_{24.58}C_{5.92}$ | 207 [19] | 263 | 27 |
| 11 | $Mo_{40}Nb_{10}V_{20}Ta_{20}W_{20}$ | 498 [27] | 665 | 33.5 |
| 12 | $Al_{14.4}Co_{16.9}Cr_{18.3}Fe_{16.8}Mn_{16.9}Ni_{16.7}$ | 535 [41] | 628 | 17.3 |

### 3.2. Experimental Results

The XRD pattern of $C_{0.1}Cr_3Mo_{11.9}Nb_{20}Re_{15}Ta_{30}W_{20}$ with the log intensity *y*-axis is shown in Figure 7. According to the image, each diffraction peak indicates most likely one single body-centered cubic (BCC) phase. No obviously distinguished peaks can be observed. This indicates the existence of a single-phase BCC phase in $C_{0.1}Cr_3Mo_{11.9}Nb_{20}Re_{15}Ta_{30}W_{20}$. The wavelength of the X-ray is 1.5406 Å. With the help of Bragg's law [42], described as

$$2d \cdot sin\theta = n \cdot \lambda \qquad (8)$$

where *d*, $\theta$, *n*, and $\lambda$ are interplanar spacing, half of the diffraction angle, positive integer, and X-ray wavelength, respectively. The lattice parameter was determined to be 3.21 ± 0.002 Å.

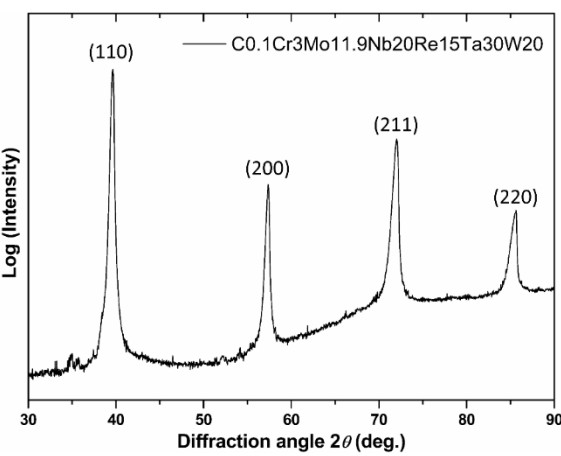

**Figure 7.** Image showing the XRD pattern of the alloy, with the *x*-axis and *y*-axis indicating the diffraction angle and log function of intensity, respectively.

The microstructure information of the as-cast $C_{0.1}Cr_3Mo_{11.9}Nb_{20}Re_{15}Ta_{30}W_{20}$ alloy is shown in Figure 8a. Two kinds of areas with differing contrast are clearly observed (lighter and darker); the lighter area is larger than the darker area, with the lighter area acting as the matrix and the darker area as the second phase. Interestingly, although two distinguishable

areas are seen, the XRD results only show a single BCC phase structure. The two areas are both BCC structures and have almost identical lattice parameters, close enough that they are beyond the detection limit of the XRD device used in this study. The shapes of the darker area are irregular, both particle-shapes (with the size of several micrometers) and strip-shapes (with a width of several micrometers and length up to several tens of micrometers) are observed. A composition test was also performed with EDS. Figure 8b exhibits the nine areas selected for the composition characterization to ensure the reliability of the results, and the composition information is listed in Table 3. Carbon elements were not listed in the table, considering the surface contamination during the polishing and storage processes. Carbon elements would be absorbed on the polished surface from the surroundings, distorting the testing accuracy. Generally, the tested composition of the as-cast alloy was consistent with the nominal composition. However, with a close observation of the table, it is discovered that the tested Cr content is considerably lower than the nominal content. This phenomenon can be explained by the following reason [43,44]: residual oxygen existed in the argon atmosphere; the as-cast alloy surface was inevitably oxidized during the synthesizing process. Cr is beneficial for oxidation resistance by generating dense $Cr_2O_3$ layer [45], and the reaction mechanism between Cr and oxygen was reported to be counter-current diffusion of these two elements, which indicates the outward diffusion of Cr during the arc melting process, reducing the Cr content inside the sample on the polished surface.

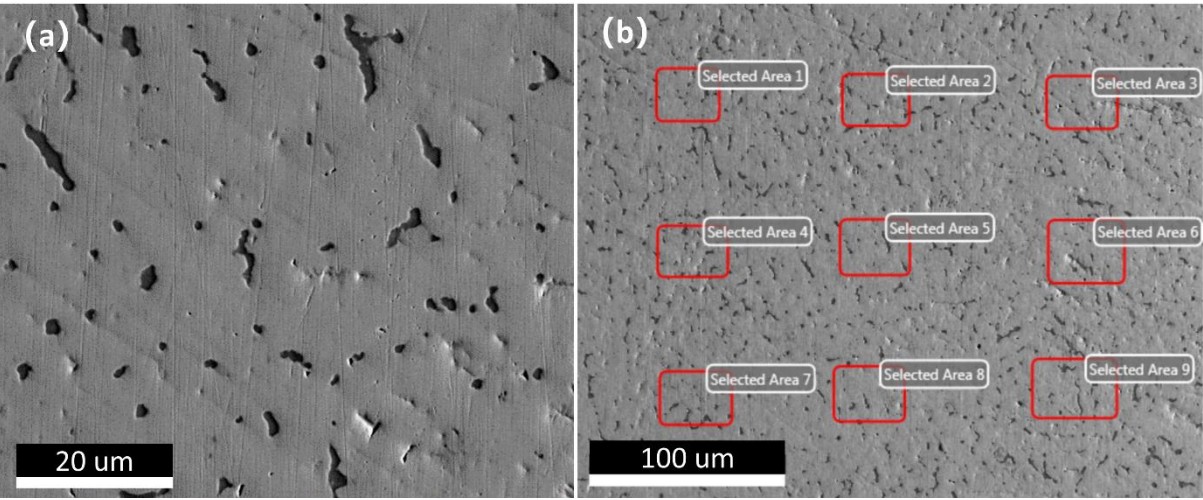

**Figure 8.** (**a**) SEM image of sample surface after polishing; (**b**) energy dispersive spectroscopy (EDS) testing areas on the sample surface.

**Table 3.** EDS testing results of the polished sample surface.

| Element | Nb | Mo | Cr | Ta | W | Re |
|---|---|---|---|---|---|---|
| Nominal (at.%) | 20.00 | 11.90 | 3.00 | 30.00 | 20.00 | 15.00 |
| Tested (at.%) | 19.31 | 8.90 | 1.06 | 31.47 | 22.66 | 16.59 |
| Standard Deviation (at.%) | 0.71 | 0.19 | 0.15 | 0.17 | 0.51 | 0.29 |

In addition, the difference of vapor pressure among the metal elements is another reason for the variation. As shown in Figure 9, the vapor pressure of the six elements from room temperature to ~2200 °C shows the following relationship [46], Cr > Mo > Nb > Re > Ta > W. Therefore, during the arc melting process, Cr and Mo were more volatile than the other elements, especially Cr.

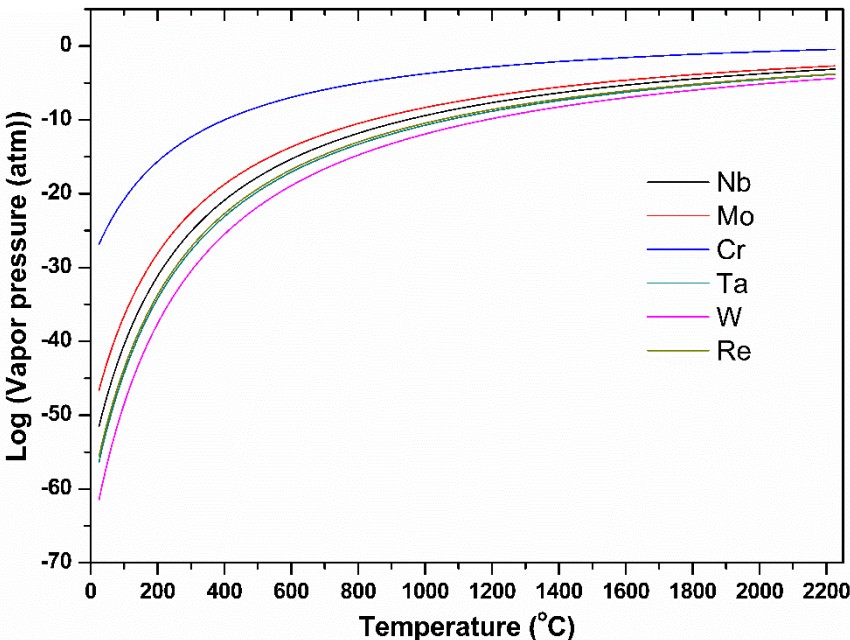

**Figure 9.** Image indicating the vapor pressure in log as a function of temperature (from room temperature to around 2200 °C).

An EDS mapping test was also performed to determine the composition distribution in the sample. All the elements except carbon were examined, and the results are displayed in Figure 10. It is clear that the differing contrasts observed in Figure 8a result from element segregation. Elements Nb and Mo are clearly present in the darker areas, while elements Cr and Ta show slightly higher contents in the darker areas. Elements W and Re exhibit opposite behaviors, and mainly exist in the lighter areas. Careful observation of Figure 10 reveals that the lighter areas are likely the dendrites, while the darker areas appear to be inter-dendrite structures. The melting points of W and Re are higher than those of Cr, Nb, Mo, and Ta. During the solidification process of arc melting, elements with higher melting points (W and Re) solidified first, and the elements with lower melting points gradually enriched at the inter-dendrite region. As a result, the segregation of elements was observed.

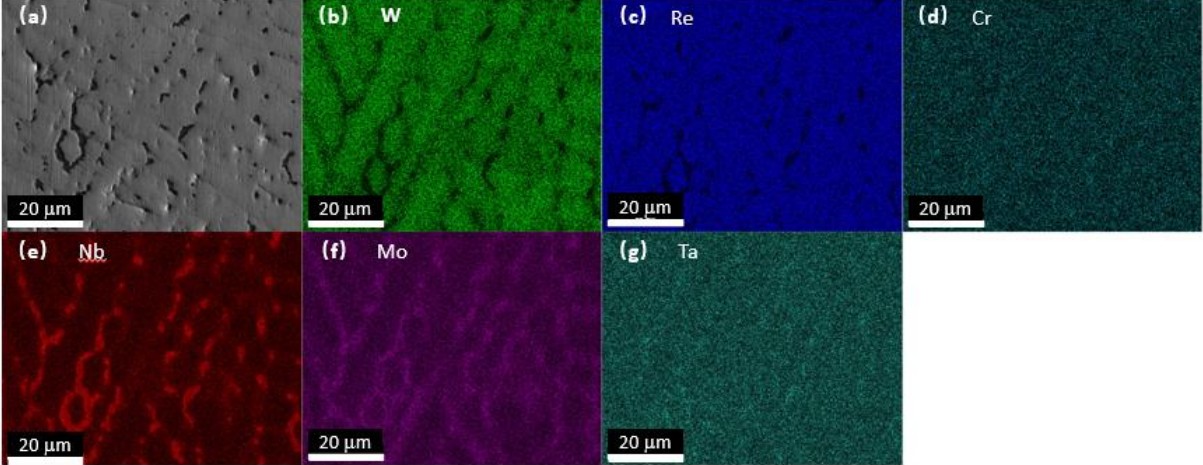

**Figure 10.** EDS mapping results of the sample $C_{0.1}Cr_3Mo_{11.9}Nb_{20}Re_{15}Ta_{30}W_{20}$. (**a**) is the SEM image showing the area for EDS mapping analysis, (**b**–**g**) indicate the distribution of W, Re, Cr, Nb, Mo and Ta elements, respectively, on the sample area.

A Vickers hardness test was carried out and the results are listed in Table 4. Based on the results, the average hardness was approximately 600 HV. Meanwhile, it was also determined that with the increase in testing load, the obtained average hardness decreases. This phenomenon is ascribed to indentation size effect [47], and similar observations were also reported by other researchers [48,49].

**Table 4.** Vickers hardness test results of $C_{0.1}Cr_3Mo_{11.9}Nb_{20}Re_{15}Ta_{30}W_{20}$.

| Load (gf) | Average Hardness (HV) | Standard Deviation (HV) |
|-----------|----------------------|-------------------------|
| 2000 | 587.10 | 21.56 |
| 500 | 595.44 | 21.35 |
| 100 | 622.60 | 13.05 |

The ML studies on mechanical and physical properties of HEAs were verified by comparing with the experiments [21–26,50–52]. The predicted Vickers hardness of $C_{0.1}Cr_3Mo_{11.9}Nb_{20}Re_{15}Ta_{30}W_{20}$ with nominal composition and experimental composition were 695 $H_V$ and 686 $H_V$, respectively. The experimentally measured Vickers hardness of $C_{0.1}Cr_3Mo_{11.9}Nb_{20}Re_{15}Ta_{30}W_{20}$ agreed with the machine learning prediction, with an error below 15%. Moreover, the calculated Vickers hardness of $C_{0.1}Cr_3Mo_{11.9}Nb_{20}Re_{15}Ta_{30}W_{20}$ was found to be 900 $H_V$ using the rule of mixture and Chen's model [53]. This predicted value of hardness had an error % of 64, which is not reliable. The hardness predictions from the current model seem very promising because they were studied with small training datasets. The close agreements between predictions and experiments validate the reliability of the current NN model. The prediction accuracy can be further improved with sufficiently larger datasets. A similar neural NN model could be used in predicting properties of future RHEAs, such as yield strength, ductility, and tensile strength, etc.

## 4. Conclusions

In this study, an NN model consisting of five hidden layers was introduced and the hardness of the novel alloy $C_{0.1}Cr_3Mo_{11.9}Nb_{20}Re_{15}Ta_{30}W_{20}$ was predicted. The microstructure study showed a single BCC phase existing in $C_{0.1}Cr_3Mo_{11.9}Nb_{20}Re_{15}Ta_{30}W_{20}$. The predicted hardness of $C_{0.1}Cr_3Mo_{11.9}Nb_{20}Re_{15}Ta_{30}W_{20}$ by the NN model was 695 HV, which agreed with the experiment. The current NN method will allow researchers to synthesize virtual RHEAs to achieve the expected hardness without any experimental trial and error method. Therefore, provides a promising scope in accelerating RHEAs designs with the desired hardness.

**Author Contributions:** Conceptualization, U.B., C.Z. (Congyan Zhang) and S.Y.; methodology, U.B. and A.A; software, U.B; C.Z. (Congyuan Zeng), A.A.; validation, C.Z. (Congyan Zhang), S.G. and S.Y.; formal analysis, C.Z. (Congyuan Zeng), and S.G; investigation, U.B., and C.Z. (Congyan Zhang); resources, S.Y.; data curation, A.A., and U.B.; writing—original draft preparation, U.B., C.Z. (Congyuan Zeng) and S.Y; writing—review and editing, S.G. and S.Y; visualization, S.Y.; supervision, S.Y.; project administration, S.Y.; funding acquisition, S.Y. All authors have read and agreed to the published version of the manuscript.

**Funding:** This research was partially supported by the NSF EPSCoR CIMM project under Award No. 1541079, DOE award No. DE-NA0003979, and DoD support under contract No. W911NF1910005. CZ(Congyuan Zeng) and SG are supported by the US National Science Foundation under grant number OIA-1946231 and the Louisiana Board of Regent for the Louisiana Material Design Alliance (LAMDA). The computational simulations were supported by the Louisiana Optical Network Infrastructure (LONI) with the supercomputer allocation loni_mat_bio15 and 16.

**Institutional Review Board Statement:** Not applicable.

**Informed Consent Statement:** Not applicable.

**Data Availability Statement:** All data and codes used in this study will be made available upon the reasonable request to the corresponding authors.

**Conflicts of Interest:** The authors declare no conflict of interest.

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
