# Peer review of "Deep Learning-Based Hardness Prediction of Novel Refractory High-Entropy Alloys with Experimental Validation"

_crystals, doi:10.3390/cryst11010046_

Round 1

Reviewer 1 Report

In this work, the authors designed and trained a neural network (NN) to predict the hardness of refractory high-entropy alloys. The NN is used to predict the hardness of a novel alloy C0.1Cr3Mo11.9Nb20Re15Ta30W20, which was synthesized in laboratory. XRD and EDS experiments are used to investigate the crystal structure and the composition of the sample. The measured value of the hardness is found to be in fair agreement with the NN prediction.

Machine learning techniques have been already used to predict properties of RHEAs. The present paper adopts the same technology to predict a specific mechanical property.

The authors successfully identified 14 features that enabled them to train a NN model achieving relatively good performance. However, the reduced size of the training data set, as pointed out by the authors themselves, at the moment does not allow to perform a more thorough assessment of the model.

I have the following comments/questions that I'd like the authors to address:

1) The authors should report the final list of 14 features selected for training.

2) The authors should comment about the criterion used to choose the number of hidden layers and the value of k=20 for the k-fold cross validation. 

3) It will be useful for the reader if the standard deviation of the NN predictions was reported directly in table 2 as well as in Fig.4, using error bars. Similarly, the authors should consider reporting shaded areas in Fig.3, indicating the standard deviation of the loss as obtained from the 20-fold cross validation.

4) For direct comparison, it will be useful to report in table 2 also the (average) experimental value obtained for the new alloy.   

5) The experimental results show that the relative concentration of the elements in the sample somewhat deviates from the nominal one. What is the NN prediction at the experimental concentrations?

6) There is a typo in equation 1. Also the definition of 'R' is missing in the text.

7) There is a typo in equation 5.

8) From line 93-94, it seems that all the three features: VEC, Tm and B have been removed. The reason is not clear. At line 97, it is not clear which are the 7 features that have been removed.

9) The authors should provide references for the algorithms adopted during features selection and training.

10) The protocol adopted to obtain the final NN predictions should be explained more clearly. Which model has been adopted? Or they represent an average over the 360 x 20 trained models ? Or just over 20 trained models, using a single set of hyperparameters? In the latter case, the author should comment on how the final hyperparameters were selected.

11) Overall, the clarity of the presentation can be improved.

12) The measured value of the hardness depends on the applied normal load. Are the experimental values of the hardness in the training set consistent with each other? This aspect needs to be addressed.

13) Figure 2 should report all the 128 predictions of the NN model.

14) Which are the closest alloys in the training set to the new one, in terms of elements concentration? And what are their hardness?

15) For completeness, Figure 3 should also report the curve of the loss on the test dataset (maybe as a second panel).

16) At line 187 the authors write: "which is consistent with our prediction from empirical parameter calculation". It is not clear to which prediction they are referring to.

17) Units of measure are missing in figure 1.

18) For clarity, the authors should explicitly report the number of samples in the training, validation and test data set.

Author Response

Thank you very much for your work concerning our paper.

Reviewer 2 Report

The authors have presented a study in which they use a Neural Network to predict the Vickers hardness in a refractory high-entropy alloy. The authors first predict the hardness of the RHEA by using information about the component parts and alloys into the NN. Subsequently, the authors produce an alloy with a nominal composition equivalent to the predicted RHEA to determine the actual hardness of the RHEA. The arc-melted RHEA is characterized with SEM and the authors find that while there is some chemical segregation, their predicted hardness value falls within 11% of the actual hardness value.

I have marked up the PDF, and it is included with this review, but I have the following comments as well:

  1. In the beginning of the manuscript, the authors use the word “features” without explicitly defining what they are referring to with the word “features.” I would ask that the authors be more explicit, or choose a more descriptive word.
  2. Please describe how you validated the input data for the NN. Did you compare multiple literature sources?
  3. Please describe what method you used for EDS quantification. Using the quantification button on EDS software is notoriously error prone. I won’t require you to run a standard-based quantification, but you should a) state how the quantification was performed, and b) offer caveats related to the values presented by the quantification method.
  4. You state Cr migrated to the surface during arc melting. Did you find an enriched layer of Cr at the surface of the arc melted ingot?
  5. You use the NN to predict the hardness of the alloy with the nominal composition. However, your EDS results clearly demonstrate that neither the matrix or the inclusions are this nominal composition. I think there is merit in this manuscript moving forward as is, but in the future, I would recommend doing the work in the opposite order. That is to say, you should produce a RHEA, measure the actual bulk composition and then based your NN predictions on the actual measured bulk composition. This would eliminate any issues with multiphase segregation and/or issues like the suspect Cr segregation to the surface.
  6. You give the average hardness as measured, and compare it to the prediction from the NN as an 11% error. However, there’s really no indication whether this is a good or bad estimate. Is there another way you can estimate hardness of this RHEA so that you can compare to the NN results? Even a simple rule of mixtures of the constituent parts would provide some comparison of how well the NN performed. For all we know, 11% could be a terrible estimate compared to another estimation method.

Author Response

(The authors gave the same response as above.)

Reviewer 3 Report

The article reports on a neural network model utilized for the prediction of the hardness of a high entropy alloy. This is a very interesting paper that ultimately should be published because the prediction of HEAs properties is a hot topic in the material science community and novel approaches are highly required. At present, however, the manuscript requires re-working because there are several important points that have to be clarified. Prior to publication, the paper should be completed and modified according to the following comments.

Comments and recommendations

Overall major comment: The authors consider five basic features for NN training: configuration entropy, the bulk modulus B, the shear modulus G, the valence electron concentration VEC, and the melting temperature. However, it is not clear why the authors consider configuration entropy as a parameter for training since configuration entropy, in contrast to four other parameters, has no effect on mechanical properties of HEAs, and on hardness in particular. It is worth noting here that atomic size difference can be considered as a much better parameter that has an influence on hardness because the main strengthening mechanism in HEAs is solid-solution strengthening which depends on severe lattice distortion of the alloys. It has been demonstrated in the literature that the atomic size difference scales with the mechanical properties of the HEAs. If the authors consider the configuration entropy as a parameter for predicting hardness and training NN, they should explain it in the manuscript and provide corresponding references. The reference [26] does not provide any information regarding the mentioned problem. However, the authors should refer to this article when considering the atomic size difference as a parameter for training the NN. Summarizing, I would strongly recommend considering the atomic size difference in training the NN.

Additionally, the authors compare their predictions with experimental results for the two-phase HEA. It is, therefore, worth explaining in the manuscript whether the NN takes into account multi-phase HEAs.

Specific comments on the results:

  1.  
  • Lines 31-33. It is worth to mention in the list of doping elements Oxygen, Boron and Nitrogen also because HEAs doped with these elements demonstrated improvement of the mechanical properties. See the following references for details: 10.1038/s41586-018-0685-y, 10.1016/j.actamat.2018.04.004
  • Lines 33 -34. What are the units of C concentration here? Please, clarify.
  • Lines 35-36. Carbon is not a metallic element. Please, re-write to be clear.
  • High entropy refractory metal nitrides can also have very high hardness when the metal to nitrogen ratio is 1:1. See 10.1016/j.compositesb.2018.04.015
  1.  
  • 4. It is worth to mention in the text and on the figure, for which HEA compositions the red star symbols are plotted. Moreover, it is worth to add error bars for the experimental data on this plot.
  • Table 2. It is not clear what the “actual hardness” means here. Is it experimental hardness or theoretical hardness? Please, re-write to be clear. Errors (standard deviation) of the “actual hardness” should be provided. The experimental hardness of the studied HEA should be also provided.
  • It might be useful to present the XRD pattern with the intensity axis in log that would help to distinguish the phases. There is a slight broadening of the peaks visible on the pattern that can be an indication of the second bcc phase with a slightly different lattice parameter.
  •  

The manuscript contains very valuable results, but they should be analyzed in more detail and interpreted correctly. Therefore, a major revision is necessary.

Author Response

(The authors gave the same response as above.)

Round 2

Reviewer 1 Report

See attached pdf.

Author Response

Thank you very much for your time and work concerning our paper.

Reviewer 3 Report

The manuscript has been improved according to my comments and recommendations. Now I can recommend publishing the manuscript in the journal Crystals. 

Author Response

We thank you for considering our manuscript for publication in the journal of Crystals